# Molecular Dynamics-Based Comparative Analysis of Chondroitin and Dermatan Sulfates

**DOI:** 10.3390/biom13020247

**Published:** 2023-01-28

**Authors:** Marta Pągielska, Sergey A. Samsonov

**Affiliations:** Faculty of Chemistry, University of Gdansk, ul. Wita Stwosza 63, 80-308 Gdańsk, Poland

**Keywords:** modelling glycosaminoglycans, chondroitin sulfate, dermatan sulfate, glycosaminoglycan sulfation code, molecular dynamics, conformational analysis, explicit solvent simulations, GLYCAM06

## Abstract

Glycosaminoglycans (GAGs) are a class of linear anionic periodic polysaccharides containing disaccharide repetitive units. These molecules interact with a variety of proteins in the extracellular matrix and so participate in biochemically crucial processes such as cell signalling affecting tissue regeneration as well as the onset of cancer, Alzheimer’s or Parkinson’s diseases. Due to their flexibility, periodicity and chemical heterogeneity, often termed “sulfation code”, GAGs are challenging molecules both for experiments and computation. One of the key questions in the GAG research is the specificity of their intermolecular interactions. In this study, we make a step forward to deciphering the “sulfation code” of chondroitin sulfates-4,6 (CS4, CS6, where the numbers correspond to the position of sulfation in NAcGal residue) and dermatan sulfate (DS), which is different from CSs by the presence of IdoA acid instead of GlcA. We rigorously investigate two sets of these GAGs in dimeric, tetrameric and hexameric forms with molecular dynamics-based descriptors. Our data clearly suggest that CS4, CS6 and DS are substantially different in terms of their structural, conformational and dynamic properties, which contributes to the understanding of how these molecules can be different when they bind proteins, which could have practical implications for the GAG-based drug design strategies in the regenerative medicine.

## 1. Introduction

Glycososaminoglycans (GAGs) are a class of linear anionic periodic polysaccharides that are made up of repetitive disaccharide building blocks containing a uronic acid (glucuronic, GlcA or iduronic, IdoA) and a hexosamine (N-Acetylglycosamide, GlcNAc or N-Acetylgalactososamide GalNAc) [1]. GAGs are located in the extracellular matrix of the cell, where they participate in many key biochemical processes such as angiogenesis, anticoagulation, cellular communication and adhesion [2,3]. Their involvement in these processes is mediated through direct interactions with diverse protein targets such as collagens [4], chemokines [5] and growth factors [6,7]. All this makes GAGs to be essential players in a number of diseases and disorders including cancer [8], Alzheimer’s [9] and Parkinson’s disease [10], autoimmune diseases [11] and arthritis [12]. Therefore, GAGs are very promising potential molecular targets for novel regenerative medicine strategies [13,14,15]. Chemically, GAGs are immensely heterogeneous. Depending on their disaccharide unit composition, glycosidic linkage and sulfation pattern, they are classified into several groups: hyaluronic acid (HA), chondroitin sulfate/dermatan sulfate (CS/DS), heparin/heparan sulfate (HP/HS) and keratan sulfate (KS). Altogether, 202 different GAG disaccharide variants in mammals are known [3]. Such heterogeneity, which could be also present within the same GAG chain, as well as GAG’s high flexibility and periodicity, renders these molecules profoundly challenging to analyse using the experimental techniques only [16]. Therefore, computational approaches could be particularly efficient in the GAG research [17,18]. Recently, many interdisciplinary studies proved that a combination of the experimental and theoretical approaches could be especially promising in studying biomolecular systems containing GAGs [19,20]. At the same time, for computational researchers, there are still many challenges related to the physico-chemical properties of GAGs to overcome. They include their anionic nature, which makes it essential to use appropriate treatment of the electrostatics, ions and solvent, which is much more abundant in protein-GAG interfaces than in the complexes of proteins with other classes of biomolecules [21]. GAGs periodicity can result in multipose binding, in which several protein-GAG complex configurations can have similar free binding energies and, therefore, co-exist [22]. Finally, one of the key challenges in understanding GAG molecular interactions relates to deciphering the “sulfation code” [23], which should assist in the explanation and prediction of GAG specificity [24,25]. A recent work of Holmes et al. can be considered as a breakthrough in terms of understanding the “sulfation code” [26]. In this work, a number of molecular descriptors of several HS variants were analysed with the molecular dynamics (MD) approach, and it was shown that a combination of these partially interdependent MD-derived parameters determine the conformational behaviour and binding propensities of the GAGs, while the data about the GAG sequence alone are not sufficient to improve the knowledge on these complex molecules in terms of the “sulfation code”.

In the present study, we aim to contribute to the comprehension of the “sulfation code” by applying MD-based analysis, which allows for essential advances in understanding GAG properties [27], to CS and DS molecules. CS and DS are very similar GAG molecules made up of GalNAc(β1→4)GlcA(β1→3) and GalNAc(β1→4)IdoA(β1→3) disaccharide units, respectively [28]. CS and DS chains are usually composed of 40 to 100 disaccharide units [29]. CS can be found in the cartilage and is involved in the bone resorption process [30], while DS can be found both in the skin, blood vessels and lungs and has antithrombotic activity [31]. When sulfated in the 4th of the 6th position of GalNAc, CS has a charge of −2 per disaccharide unit (CS4 and CS6, respectively). DS has predominantly sulfation in the 4th position and its disaccharide unit has also a net charge of −2 [32]. Sometimes, GlcA and IdoA can undergo epimerisation, and then a GAG chain has both CS and DS parts. Despite their similarities, CS and DS have very distinct protein binding properties: in particular, there is experimental evidence so far suggesting that if there are complexes with DS, the same proteins also bind other GAGs, while there are many indications of the CS binding specificity, meaning that some protein can bind only CS but not DS [3]. It was also shown, both with NMR and molecular dynamics (MD)-base techniques that CS4 and CS6 can bind with significantly different affinities to the same proteins [33] suggesting that it is not exclusively a net charge that drives these interactions. MD simulations have also proved to be successful in reproducing experimental data for unbound CS oligosaccharides. Already in 1999, Kaufmann et al. performed 4 ns MD simulations of the CS4 tetrasaccharide to characterise its glycosidic linkage conformational space, hydration properties and H-bonding [34]. It was shown that in comparison to non-sulfated oligosaccharides, CS is highly hydrated thanks to its negative charge. Interestingly, this hydration partially remains high also when they bind to proteins [21]. Samantray et al. characterised the dynamics of several GAG disaccharides including CS4 and CS6 and established the dependence of their behaviour on the salt presence and type [35]. Guvench’s group constructed atomic models of CS/DS oligosaccharides, rigorously analysed their conformational space and established the dependence of their conformational ensembles on the interactions with Ca${}^{2}{}^{+}$ and their sulfation pattern [36,37,38,39,40]. Recently, MD simulations with CHARMM and GLYCAM06 force fields were used to characterise all possible disaccharide variants of CS in terms of their free energy landscape [41,42]. In the latter publication, the authors performed a rigorous analysis of the disaccharide torsional space, intra- and intermolecular H-bonds, bridging water molecules and principal components of movements. They conclude that the observed distinct conformational dynamism in different CS disaccharides is the reason that unique electrostatic surfaces exist that could be a key for protein recognition.

In this work, we compare the properties of CS4, CS6 and DS of different lengths (di-, tetra- and hexasaccharides—dp2, dp4 and dp6, where dp stays for the degree of polymerisation) in two chain variants (Set 1 and Set 2) differing from each other by the type of the residue at the reducing and non-reducing ends of the oligosaccharides (Figure 1).

In total, 18 oligosaccharides are analysed and compared to dissect the effect of the sulfation and the type of uronic acid on the dynamic and conformational properties of these GAGs, which should, in turn, determine the differences in their binding specificity. In particular, flexibility defined in terms of the atomic fluctuations, the radius of gyration, end-to-end distance, molecular volume, interactions with the ions and solvent, internal hydrogen bonds, glycosidic linkage and ring puckering conformations have been used as molecular descriptors similarly to other GAG conformational studies [26]. We find that the differences between CS4, CS6 and DS can be established by considering a combination of the analysed dynamic parameters. Our data are in line with the results obtained for 3-O-sulfated HS variants by Holmes et al. suggesting that the GAG “sulfation code” is a concept which understanding should be established based on the data obtained from the dynamics of these systems [26]. In these terms, our work adds to the general knowledge about the “sulfation code”, which could be of high relevance for the novel approaches in the GAG-related drug development for a number of diseases where these molecules are mediators of the underlying molecular processes.

## 2. Materials and Methods

### 2.1. Structures

The GAG structures used in this study were built from the previously modelled structures of the corresponding GAG molecules [33] originally obtained from the PDB (PDB IDs: 1CS4 and 1HM2). In total, 18 different GAG oligomers were built: dp2, dp4 and dp6 of CS4, CS6 and DS included in Set 1 and Set 2, depending on if they contain a NAcGal residue or a uronic acid residue at the non-reducing end, respectively.

### 2.2. Molecular Dynamics Simulations

Molecular dynamics simulations were performed in AMBER20 package [43]. GAG oligosaccharides were solvated in TIP3P water [44] in an octahedral periodic box with the minimum distance between solute and box edge of 10.0 Å and neutralised with counterions (Na^+^). GLYCAM06 forcefield parameters [45] with the previously described sulfate charges compatible with GLYCAM06 force field [46] were used. Two energy minimisation steps were performed (first consisted of 1.5 × 10${}^{3}$ steepest descent cycles and 10${}^{3}$ conjugate gradient cycles with harmonic force restraints of 100 kcal mol${}^{-1}$ Å${}^{-2}$ on solute atoms, followed by 6 × 10${}^{3}$ steepest descent cycles and 3 × 10${}^{3}$ conjugate gradient cycles without restraints). Then, the system was heated up to 300 K for 10 ps with harmonic force restraints of 100 kcal mol${}^{-1}$ Å${}^{-2}$ on solute atoms, followed by equilibration for 100 ps at 300 K and 10${}^{5}$ Pa in the isothermal isobaric ensemble (NPT) for the explicit solvent simulation. A productive MD run was performed in an NPT ensemble in the explicit solvent simulations for 1 μs. The SHAKE algorithm, 2 fs time integration step, 8 Å cutoff for nonbonded interactions, and the Particle Mesh Ewald method were used.

### 2.3. Analysis of Molecular Dynamics Trajectories

The trajectories were analysed using the CPPTRAJ module from AMBER package [43] and in-house Python 3.8.9 [47] and bash scripts. The following particular molecular descriptors and parameters were obtained and analysed.

–RMSD (root mean squared deviation) was calculated for all atoms of the GAG molecule and describes the changes in the molecular structure in time with the reference to its initial structure.–R${}_{gyr}$ (radius of gyration) was calculated for all atoms as the root mean squared difference between their coordinates and their geometric centre. This shape of molecular parameter characterises how elongated the molecule is.–EED (end-to-end distance) was defined as a distance between the reducing end O1 atom and non-reducing end glycosidic linkage oxygen atom and, therefore, corresponds to the length of the molecule.–Volume of the molecule was defined as the minimum volume enclosing ellipsoid for each molecule often used in the analysis of GAG conformations [48].–Glycosidic linkage dihedral angles were defined by O5-C1-O1-O4′ and C1-O1-O4′-C5′ for GalNAc(β1→4)GlcA/IdoA (linkage 1) and by O5-C1-O1-O3′ and C1-O1-O3′-C4′ for GlcA/IdoA(β1→3)GalNAc (linkage 2), respectively.–Ring puckering was defined by Cremer-Pople dihedral angles [49]. Populations of ${}^{1}$C${}_{4}$, ${}^{4}$C${}_{1}$, ${}^{2}$S${}_{O}$ and ${}^{1}$S${}_{3}$ conformations have been particularly analysed. The dihedral angle value ranges considered for the definition of each of these ring conformations were defined as in our previous work [50].–Electrostatic repulsion was described in terms of the Linear Interaction Energy electrostatic component calculated for the atoms making up carboxyl (AMBER mask: C6, O6A, O6B) and sulfate (AMBER mask: S, O1S, O2S, O3S) groups.–Interactions with counterions were defined for the carboxyl and sulfate groups by distance (between the ion and the center of mass of the group) cut-offs of 4.0 Å and 4.2 Å, respectively.–Hydrogen bonding was defined by default cut-offs in CPPTRAJ: the distance between heavy atoms of less than 3.0 Å and an angle greater than 135° for both intramolecular and intermolecular hydrogen bonds.

### 2.4. Statistical Analysis/Visualisation

The statistical analysis and data visualisation were performed in R [51]. The structures were visualised using the VMD program [52].

## 3. Results and Discussion

### 3.1. Flexibility

The analysis of the RMSD values in the course of the MD simulation suggests that the trajectories for all the oligosaccharides converged already after the first nanoseconds in terms of the RMSD (Figure A1 and Figure A2). The complete convergence which would include the ring puckering conformational space of uronic acids requires, however, could be reached in the MD simulations at the μs timescale [53]. The qualitative difference between the behaviour of CS4 and CS6 in comparison to DS is that these RMSD values, calculated with the reference to the initial structure, change more frequently for the former ones while remaining longer oscillating about a certain value after a substantial change for DS. This could potentially suggest a faster exchange between different conformational states for CSs and a slower one for DS. For DS, distinguishable bimodal distributions of RSMD values could be seen (Set 1 dp4, set 2 dp2 and dp4), which could be related to the higher puckering flexibility of IdoA ring in comparison to GlcA ring (Figure 2 and Figure 3). RMSD variance in the MD simulation could be interpreted as a measure of the flexibility of the analysed oligosaccharides. The broader the distribution of the RMSD, the more flexible the molecule. A comparison of the variances presented as violin plots shows only slight differences between the oligosaccharides, where CS4 and DS are less flexible, while CS6 is the most flexible. This is also reflected in Figure A3 showing the snapshots from the trajectories for dp6 GAGs. In the MD study performed with the CHARMM force field, CS6 also revealed higher flexibility than CS4 [35]. Such differences in flexibility could be related to the longer distance between the sulfate group in the 6th position and the sugar ring in CS6 in comparison to the sulfate in the 4th position in CS4 and DS, which is reflected in RMSD values. In general, higher flexibility can potentially mean a higher variety of conformations available upon binding, which affects the entropic component in the thermodynamic pattern of the complex formation.

### 3.2. Shape

The shape of the analysed GAGs has been described in terms of R${}_{gyr}$ (Figure 4). CS6 has the most elongated structure regardless of a chosen set. Differences between the sets are visible for DS, where dp2 and dp4 have two major visually distinguishable conformational populations, while Set 1 contains a single one. For Set 2, these populations are reflected in a single conformational change during the MD simulations (Figure A4 and Figure A5). In both sets, DS has the most defined structure which corresponds to the most narrow distribution of R${}_{gyr}$ values. For CS4 and CS6 there are multiple and reproducible events of conformational changes observed in the MD trajectories. These differences of R${}_{gyr}$ behaviour in time for both CSs in comparison to DS potentially mean that the free energy barrier between two conformational states is significantly lower for the former ones. The clear ranking of R${}_{gyr}$ presents as following:

DS < CS4 < CS6 for dp4 and dp6, while for dp2 qualitatively similar trend is observed in the case of DS Set 2. As expected, for shorter oligosaccharides the differences between the sets are more substantial due to the more significant difference in their chemical structures. Similarly to the RMSD data discussed above, these results suggest principal differences between the flexibility/conformational availability in CS and DS that would lead to qualitative differences in the thermodynamics and kinetics of their binding to a protein.

### 3.3. Global Conformational Space

The global conformational space could be described as a probability function in two dimensions: EED and molecular volume, which combination was previously shown to be an effective descriptor for distinguishing GAG conformations [48]. Except for CS6 Set 1 dp2 and DS Set 2 dp2, where one additional minor conformation is sampled, all the oligosaccharide show a clear single maximum of the probability (Figure 5 and Figure 6). In some cases, this maximum is split into two along the volume coordinate (CS4 set 2 dp4, CS6 set 2 dp2 and dp4, DS set 2 dp6). Nevertheless, the topologies of all the heatmaps as well as the EED evolution in the MD simulation and its distribution are very similar independently of the GAG type (Figure A6, Figure A7, Figure A8 and Figure A9). The analysis of EED distributions for CS4 and CS6 obtained with the MD simulations with CHARMM parameters suggest that they can slightly differ depending on the presence and type of the salt in the system [35]. The lengths of the oligosaccharides are also similar, while the volumes could be ranked as follows: DS, CS4 << CS6. In contrast to the data obtained for two descriptors analysed above (RMSD and R${}_{gyr}$), the data presented here in the heatmaps do not provide any clear implication for any difference in binding of the GAGs to proteins except for the speculation about the sizes and geometries of the putative binding sites on the protein surface required to effectively accommodate these ligands.

### 3.4. Local Conformational Space

The local conformational space could be characterised by glycosidic linkage and ring puckering conformational ensembles. Two glycosidic linkages are other conformational descriptors that could be used to characterise the flexibility of the GAG chain. The broader the regions in the glycosidic heatmaps, the more flexible the linkages and the corresponding molecule as a whole. We calculated the values of dihedral angles defining glycosidic linkages of two types (linkage 1: O5–C1–O1–O4′ and C1–O1–O4′–C5′ for GalNAc(β1→4)GlcA/IdoA; linkage 2: O5−C1−O1−O3’ and C1−O1−O3’−C4’ for GlcA/IdoA(β1→3)GalNAc) and summarised them by type for each of the simulated oligosaccharides. Both types of glycosidic linkages have similar general patterns for all three oligosaccharides independently of their types and lengths (Figure 7 and Figure 8). All the observed regions contain the conformational space observed previously in the shorter (20 ns) MD simulations [33], which, in turn, accommodate the values from the PDB corresponding to the experimentally obtained bound structures of the same GAGs suggesting that in terms of glycosidic linkages, there are no significant conformational changes occurring in the process of binding to proteins. At the same time, 1 μs simulations also reveal some additional free energy minima not observed in the short simulations suggesting the existence of minor conformational populations distinguishable from the major ones. While the glycosidic linkage 1 is very similar through all the analysed oligosaccharides, the differences were found in the glycosidic linkage 2 heatmaps. For this descriptor, the regions occupied by DS are substantially more restricted than the ones of CS4 and especially of CS6 suggesting that CSs are more flexible in terms of their glycosidic linkage conformations which could be an important factor for the dynamics and energetics of their binding to proteins. This analysis of the conformational space of glycosidic linkages partially explains the observed difference in the flexibility for the analysed oligosaccharide types. These results could also support the potential difference in terms of protein recognition by these molecules.

While NAcGal residues of all the analysed GAGs mainly remain in the ${}^{4}$C${}_{1}$ conformation, and only some changes are observed in the terminal residues, the puckering of GlcA in CS4 and CS6 and IdoA in DS are found to be significantly different (Figure 9, Table A3). In CS4 the GlcA conformation is predominantly ${}^{1}$C${}_{4}$, while the fractions of ${}^{4}$C${}_{1}$ and ${}^{2}$S${}_{O}$ in CS6 are already significant, contributing with 12.5% and 7.8% to the total puckering ensemble, respectively. For IdoA the fraction of ${}^{4}$C${}_{1}$ puckering (almost 40%) is comparable with one of ${}^{1}$C${}_{4}$, (about 60%). These results allow one to see the clear effects of both the sulfation position in the adjacent NAcGal residue on the conformation of the uronic acid as well as of the uronic acid type itself. Both of these effects could have crucial meaning in the determination of the binding propensities and specificity of these GAGs. Previously, for heparan sulfate, it was also shown that the neighbouring residues affect the pucker populations of the IdoA residue [54,55]. The differences in the GlcA and IdoA puckering conformations observed in our study should be, however, interpreted with care. It is known that uronic acids in a monomeric form fluctuate between their major ring pucker conformers on the μs timescale in the simulations with GLYCAM06 force field [53], while each of our simulations had 1 μs length. Since our data present in Figure 9 are obtained without any torsional angle restraint for the rings starting in the same particular conformation, and for each of the residue types within a particular oligosaccharide type there are 12 μs of MD simulation, in total, the observed values could only qualitatively reflect the differences in the puckering conformations that are affected by the limitations of the used force field. Therefore, they cannot be understood strictly in quantitative terms. At the same time, the absence of the torsional restraints allowed us to observe that the terminal residues are substantially more flexible in terms of the puckering conformations in comparison to the residues in the middle of the oligosaccharide sequences (Table A3). In terms of potential consequences for the interactions with proteins, this higher flexibility of the IdoA ring in DS in comparison to the one of GlcA in CS means differences in the entropic pattern of binding. However, it is not clear if the entropic differences corresponding to this puckering flexibility are decisive for determining the affinity of protein-GAG interactions.

### 3.5. Intramolecular Interactions

In the work of Holmes et al., it was shown that for deciphering GAG “sulfation code” for 3-O-sulfated heparan sulfates, the internal electrostatic tension could be a useful descriptor [26]. For both sets, there is higher repulsion between the charged groups of DS in comparison to CS (Figure 10). Similarly to other descriptors, DS shows a bimodal distribution of this descriptor, and the interconversion between the corresponding two states is slow, while the descriptor distributions of CS oligosaccharides are broader (Figure A10 and Figure A11). As for the previously discussed descriptors, this can be attributed to the distinguishable puckering variability of IdoA in DS and to the higher distance between the sugar ring and the sulfate in the 6th position in comparison to the one in the 4th position. The potential implication of such differences could be a very distinct electrostatic interaction pattern, both in terms of the strength of the interactions as well as in terms of their selectivity, for these three GAG types upon the protein binding. Similarly to the glycosidic linkage analysis, these data contribute to the understanding of the origin of the flexibility differences for the analysed GAGs, which, in turn, underlie their binding propensities.

Since GAGs are negatively charged, they interact with positive ions through electrostatic attraction that could affect their conformational and binding properties [56,57,58,59]. We analysed how often each group of the analysed GAGs established contact with the counterions in the MD simulation. Despite different accessibility of the carboxyl and sulfate groups in CS4, CS6 and DS, no differences were obtained for this descriptor (Table 1, Table A4 and Table A5). Independently of the GAG type or length, a carboxyl group and sulfate groups are in contact with the counterions for 0.018–0.023 (COO${}^{-}$) and 0.013–0.015 fractions (SO${}_{3}$${}^{-}$) of the total MD simulation, respectively. The observed significantly higher propensity of COO${}^{-}$ in comparison to the SO${}_{3}$${}^{-}$ one to coordinate counterions was reported also for other GAGs [35]. For binding to the proteins, the results suggest that independently of their positions, the free energy changes related to the disruptions of the interactions between the charged groups and counterions upon protein binding are very similar and cannot be used to discern the differences between the analysed GAG types.

H-bonding in the biomolecular systems, including protein-GAG complexes, could be the key for understanding their binding specificities and affinities [60]. Furthermore, intramolecular H-bonds play a major role in understanding GAG chain structural dynamics [61]. First, we analysed the intramolecular H-bonding patterns for the three types of GAGs (Figure 11 and Figure 12). Independently of the set, the H-bonding pattern was found to be dependent on the residue type but not on the position of the respective residue within the oligosaccharide chain reflecting the periodicity of these molecules. This did not apply to dp2 oligosaccharides since for such short oligosaccharides both disaccharide residues are terminal and, therefore, have distinct physico-chemical properties in comparison to the same residue types within a longer chain. For both sets, there are clear differences in the amount of the internal H-bonds established within the GAG chain: CS4 and CS6 form more H-bonds than DS with slightly more H-bonds established within the CS6 molecule, which agrees with the data from CHARMM simulations suggesting more intramolecular contacts in CS6 in comparison to CS4 [35]. Another important observation is a pronounced difference in the patterns of the H-bonds between all three different GAG types. The accessibility of the specific GAG H-bonding acceptors could be key for both the kinetics and thermodynamics of the protein binding for these molecules, determining their differences in specificity in the complex formation. At the same time, the number of H-bonds established with the solvent molecules is very similar for all the oligosaccharides (with the exception of dp2) suggesting their similar hydration properties independently of the GAG type (Figure A12), which is in agreement with another MD study deploying CHARMM force field [35]. These data mean that whereas similar differences in terms of the hydration are expected to occur for these GAGs when they bind proteins, there is qualitatively and quantitatively distinct H-bonding intra- and intermolecular organisation for all three GAG types. The differences between DS and both CS in these terms are especially essential.

## 4. Conclusions

We applied MD-based analysis with the aim to understand the differences between CS4, CS6 and DS oligosaccharides in terms of several structural/dynamical descriptors (RMSD, R${}_{gyr}$, EED, molecular volume, glycosidic linkages and puckering conformational space, internal electrostatic tension, interactions with counterions and H-bonding propensities). The descriptors are partially interdependent and could, therefore, indicate similar trends. At the same time, some of them are able to reveal significant differences between the analysed molecules, which are indistinguishable from other descriptors. Such significant differences have been observed for the flexibility of the studied GAG types, suggesting DS to be more rigid than both CSs, while, at the same time, the puckering space of IdoA, which is one of the residues within the disaccharide repetitive unit of DS, is essentially broader than the one of GlcA, a residue in both CSs instead. The conformational ensemble of DS could be divided into two major groups of conformations, for which the exchange is relatively slow, while CSs conformations represent a single major population exchanging rapidly with several minor conformational populations. In terms of shape, DS is the most compact, while CS6 is the most elongated molecule. Electrostatic properties and H-bonding propensities for all three GAGs are essentially different: CS6 has the highest internal electrostatic tension, while DS has the least intramolecular H-bond acceptors that are available for establishing an H-bond upon binding to a GAG binding partner. These results suggest principal differences in the kinetic and thermodynamic patterns of the intermolecular interactions that can be established by the three GAG types with proteins and other biological binding partners. In particular, the in silico obtained data for these GAGs in terms of their potential protein binding specificity can be interpreted as follows. 1. The sulfation in position 6 contributes to the higher flexibility of the GAG in comparison to the sulfation in position 4 and, so could favouritise a higher diversity of its possible conformations upon binding, leading to the lower entropic loss upon the formation of a protein-GAG complex and, therefore, higher affinity. At the same time, this higher flexibility of the CS6 could also result in the binding poses being more specific than in the case of CS4 and DS. 2. There are fewer available H-bond acceptors in the unbound CS than in the DS meaning that there are more putatively specific patterns of the H-bonding for CS to be expected when bound to the protein. 3. Finally, a more flexible IdoA ring puckering conformational ensemble in DS in comparison to the GlcA one in CS could be the reason for the different entropic patterns of these GAGs in the binding. Systematic and rigorous experiments should be performed to find out which thermodynamic trends qualitatively suggested by our data, are dominant in the protein-GAG binding. Isothermal titration calorimetry could potentially be able to answer this question and verify our findings. To summarise, this study represents a systematic step towards understanding the molecular basis of the GAG “sulfation code” and its dynamic nature, which could be of high potential interest for the GAG-based drug design in the field of regenerative medicine.

## Figures and Tables

**Figure 1 biomolecules-13-00247-f001:**
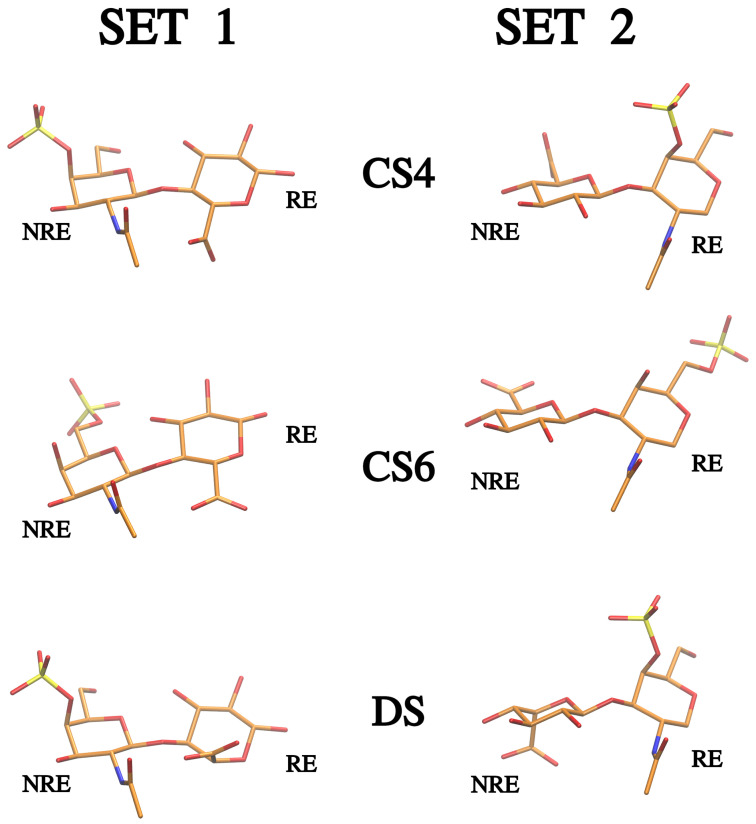
Disaccharide units of two sets of oligosaccharides used in the study. NRE and RE stay for non-reducing and reducing ends, respectively. Set 1. CS4: NRE-GalNAc(4S)(β1→4)GlcA(β1→3)-RE; CS6: NRE-GalNAc(6S)(β1→4)GlcA(β1→3)-RE; DS: NRE-GalNAc(4S)(β1→4)IdoA(β1→3)-RE. Set 2. CS4: NRE-GlcA(β1→3)GalNAc(4S)(β1→4)-RE; CS6: NRE-GlcA(β1→3)GalNAc(6S)(β1→4)-RE; DS: NRE-IdoA(β1→3)GalNAc(4S)(β1→4)-RE.

**Figure 2 biomolecules-13-00247-f002:**
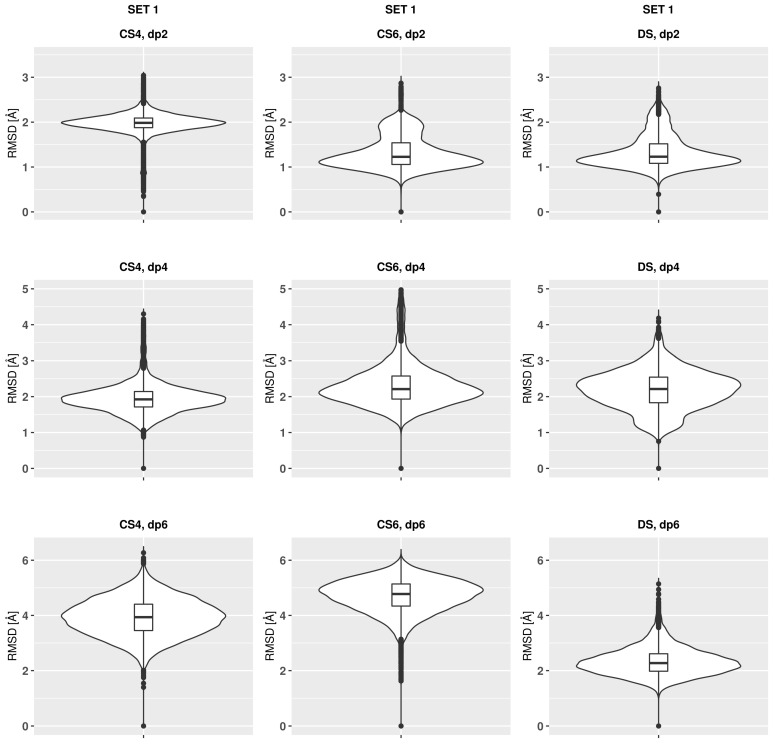
RMSD violin plots of the oligosaccharides from Set 1.

**Figure 3 biomolecules-13-00247-f003:**
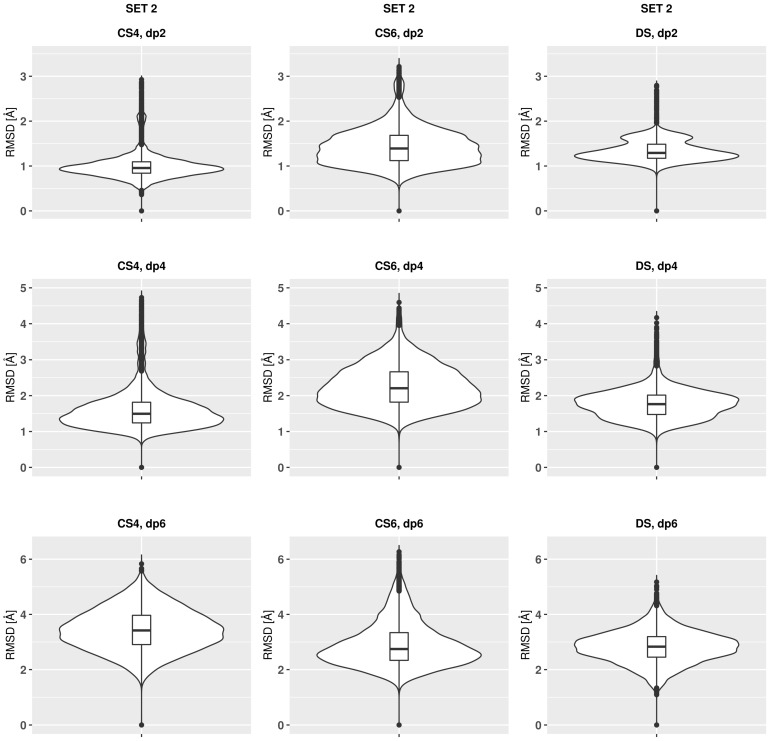
RMSD violin plots of the oligosaccharides from Set 2.

**Figure 4 biomolecules-13-00247-f004:**
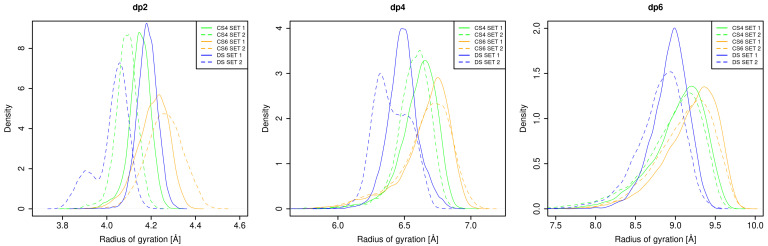
Density of probability for R${}_{gyr}$ for two sets of CS4, CS6 and DS dp2, dp4, dp6.

**Figure 5 biomolecules-13-00247-f005:**
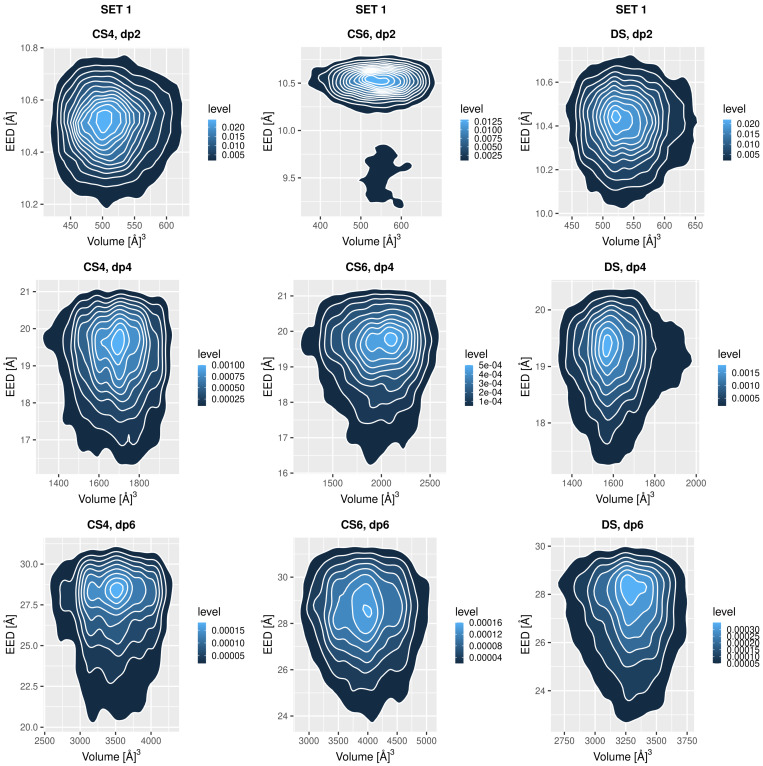
Probability heatmap of the oligosaccharides from Set 1 as a function of EED and volume coordinates.

**Figure 6 biomolecules-13-00247-f006:**
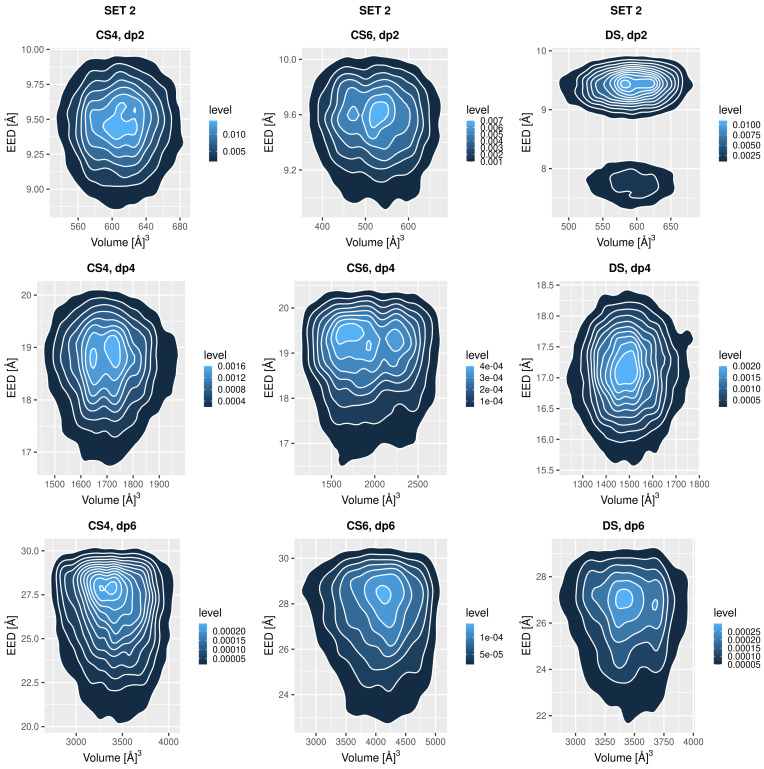
Probability heatmap of the oligosaccharides from Set 2 as a function of EED and volume coordinates.

**Figure 7 biomolecules-13-00247-f007:**
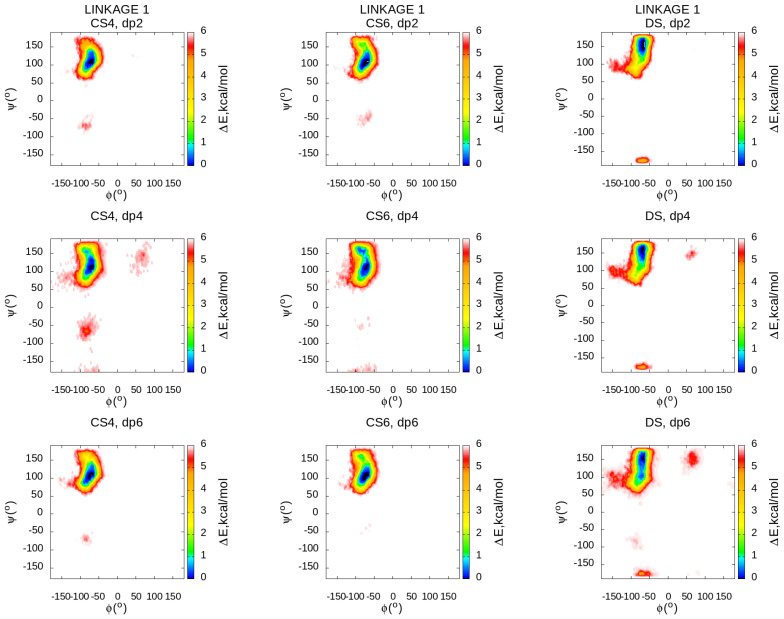
Glycosidic linkage 1 heatmaps for ϕ and ψ dihedral angles. The linkage is defined by O5–C1–O1–O4′ and C1–O1–O4′–C5′ for GalNAc(β1→4)GlcA/IdoA. The data shown here are obtained from all linkages of this type within each molecule.

**Figure 8 biomolecules-13-00247-f008:**
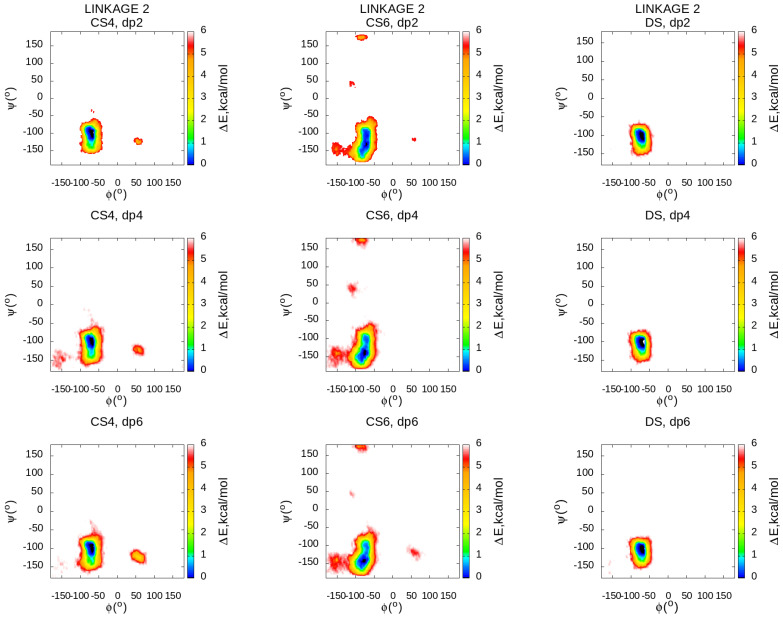
Glycosidic linkage 2 heatmaps for ϕ and ψ dihedral angles. The linkage is defined by O5–C1–O1–O3′ and C1–O1–O3′–C4′ for GlcA/IdoA(β1→3)GalNAc. The data shown here are obtained from all linkages of this type within each molecule.

**Figure 9 biomolecules-13-00247-f009:**
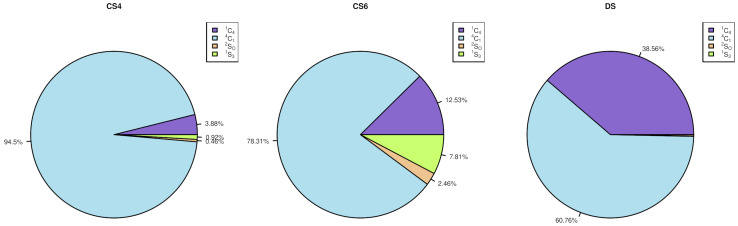
GlcA/IdoA puckering in the studied oligosaccharides. The data for all residues of the same type for all oligosaccharides are summarised.

**Figure 10 biomolecules-13-00247-f010:**
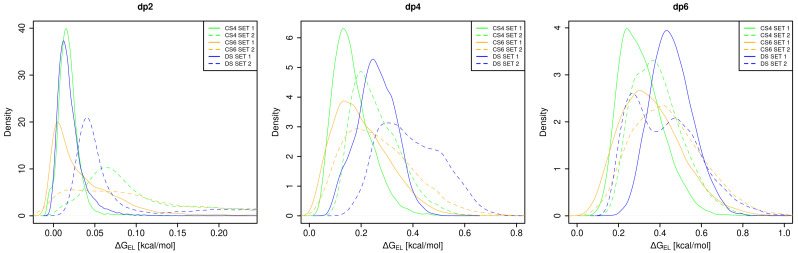
Density of probability for repulsive electrostatic energies.

**Figure 11 biomolecules-13-00247-f011:**
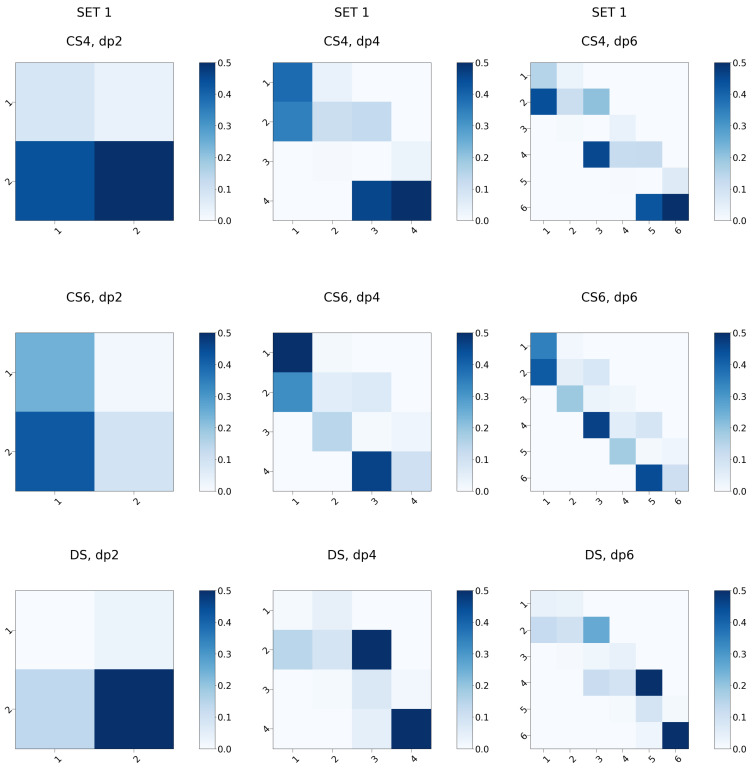
H-bond heatmaps for the oligosaccharides in Set 1. x- and y-axes correspond to the oligosaccharide residue numbers, the heatmap colour intensity reflects the summed fraction of the H-bonds established within each oligosaccharide residue (on the diagonal) or between different oligosaccharide residues.

**Figure 12 biomolecules-13-00247-f012:**
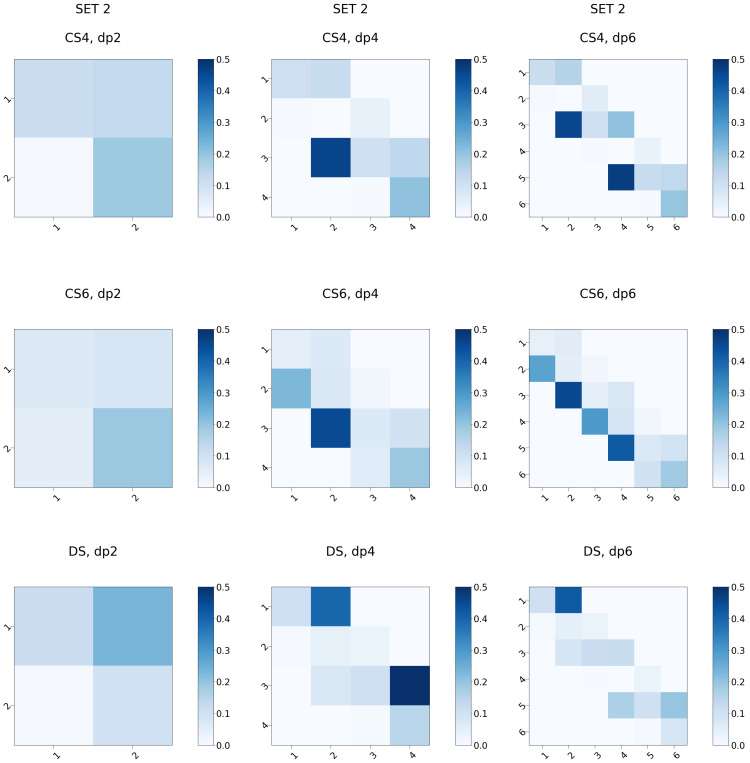
H-bond heatmaps for the oligosaccharides in Set 2. x- and y-axes correspond to the oligosaccharide residue numbers, the heatmap colour intensity reflects the summed fraction of the H-bonds established within each oligosaccharide residue (on the diagonal) or between different oligosaccharide residues.

**Table 1 biomolecules-13-00247-t001:** Fraction of contacts with counterions established by the chemical group in the analysed oligosaccharides.

GAG	COO${}^{-}$	SO${}_{3}$${}^{-}$
CS4	0.01860 ± 0.0105	0.0137 ± 0.0042
CS6	0.0189 ± 0.0134	0.0145 ± 0.0050
DS	0.0231 ± 0.0152	0.0139 ± 0.0045

## Data Availability

Not applicable.

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
