# Peer review of "Molecular Dynamics-Based Comparative Analysis of Chondroitin and Dermatan Sulfates"

_biomolecules, 2023, doi:10.3390/biom13020247_

Round 1

Reviewer 1 Report

The paper by Pagielska and Samsonov reports an MD study of CS and DS di-, tetra- and hexa- saccharides using various molecular descriptors such as Rgyr, MVEE, EED, LIE, H-bonding, etc. The study concludes that DS is more conformationally diverse and compact than CSs; CS6 has higher internal electro-repulsion; and many other properties are relative similar between the three classes of GAG oligosaccharides. In combination, despite the similarity of basic structures, the three classes display significant different conformational space, alluding to the possibility that there exists a ‘sulfation code’ at a dynamic level. For example, differences in elongation between CS and DS oligosaccharides results in the former showing relatively less electrostatic repulsion energy, which has consequences on dynamic equilibrium. The work is conducted well, although not adequately described resulting in lack of clarity with regard to presentation. In some cases, the paper’s results also remain unclear. Following suggestions should be implemented before acceptance.

1)      The authors seem to have missed a key paper that was published in 2022 in Biomolecules (vol. 12, page 77). This paper studies MD of 16 CS disaccharides (which includes DS too) using MD followed by various types of analysis to conclude “distinct conformational dynamism that offers high possibility of unique electrostatic surfaces for protein recognition” are exhibited by the studied disaccharides. This conclusion appears to be very similar to that made in the current paper. The authors should comment on the correspondence, or lack thereof, between the 16 CS disaccharide work and the current work with 8 oligosaccharides.

2)      Two additional works that the authors seem to have missed or avoided are the MD of CS tetrasaccharide (PMID: 10515047) and energy landscape analysis of CS in the presence of cations (PMID: 34768961). Both appear to be relevant to this work and authors should comment on the correspondence, or lack thereof, between these prior works and their current work.

3)      Figure 1: Legend should be more descriptive. Sequence name starting with the label ‘NRE’ and ending with the label ‘RE’ would be helpful to those not too familiar with GAGs. Also, the orientation should be same (not all atoms are visible in each image which leads to some confusion).

4)      Figure A2: Is there a reason why puckering was not restricted using torsional angle restraints? IdoA has been shown to fluctuate between its major conformers on the microsecond timescale (PMID: 20809637). So allowing full ring flexibility on a timescale less than that could be conceived as unrealistic.

5)      For Figure 7, provide chemical structures of the linkages. dp6 has 5 linkages, yet only 'linkage 1' and 'linkage 2' are shown. What is meant by linkage 1 and linkage 2? How do these relate to those of set 1 or set 2?

6)      In Figure 11, intramolecular hydrogen bonding plots are confusing. What do the x and y axes represent? For example, in CS4 dp2, what is the difference between a correlation between 1 and 2, and then between 2 and 2?

7)      In general, this reviewer found it difficult to read the x- and y-axes labels and titles in multiple figures (e.g., 2, 3, 4, 5, 6, ….). Similarly, the figures in appendix are also not very readable. These should be greatly improved. A good way to check for format is to print and try to read from a reasonable distance.

8)      Line 83: No citations are provided for molecular descriptors. Please add.

9)      Line 96: Better to add PDB IDs rather than just stating PDB.

10)   Lines 86 – 90: Not only is this sentence too long but it is also incomplete. Break it down and clarify.

11)   Lines 159 – 160: Offering a pictorial view of these observations in the form of movie (supplementary?) would be very helpful. How do the authors define conformational flexibility is not clear. It cannot really be based on RMSD alone.

12)   The authors state (pg 11, ln 206) that a “conformational selection rather than an induced fit mechanism of binding to proteins would be realized”. It is not clear how this conclusion was derived. An explanation should be provided, especially because no analysis has been done in the bound form. In fact, the authors could check multiple computational and non-computational papers published on CS – protein systems and assess whether their hypothesis is correct. Alternatively, the authors could revise it or delete the hypothesis.

13)   The authors state that convergence was achieved in first few nanoseconds and yet the authors performed simulations for 1 microsecond. Having no restraints on IdoA residues is usually a problem. How were they treated and how was puckering analysed and ascertained to not deviate from what is known?

14)   Figure A2 legend: Should the ratio be 1C4, 4C1, 2SO and then 1S3, instead of 4C1, 1C4, 2SO, and 1S3?

15)   Typographical or grammatical errors should be removed during re-submission. A) Line 1: GAGs is a class (should be ‘are’); B) Line 8: ‘by IdoA acid’ should be ‘in the presence of IdoA instead of ..’; C) Line 16: Same as Line 1; D) Line 57: ‘anvances’ should be ‘advances’; D) Line 63: Cite a reference instead of ‘?’; E) Line 64: ‘predominanly’ should be ‘predominantly’; F) Line 75: fix the charge as superscript; G) Line 190: ‘Oligosacchairide’ should be ‘oligosaccharide’.

Author Response

REVIEWER 1

POINT 1. The authors seem to have missed a key paper that was published in 2022 in Biomolecules (vol. 12, page 77). This paper studies MD of 16 CS disaccharides (which includes DS too) using MD followed by various types of analysis to conclude “distinct conformational dynamism that offers high possibility of unique electrostatic surfaces for protein recognition” are exhibited by the studied disaccharides. This conclusion appears to be very similar to that made in the current paper. The authors should comment on the correspondence, or lack thereof, between the 16 CS disaccharide work and the current work with 8 oligosaccharides.

ANSWER. We thank the Reviewer for this important work on CS that we missed. It has been added to the references (42) and its results have been cited appropriately in the revised manuscript.

POINT 2. Two additional works that the authors seem to have missed or avoided are the MD of CS tetrasaccharide (PMID: 10515047) and energy landscape analysis of CS in the presence of cations (PMID: 34768961). Both appear to be relevant to this work and authors should comment on the correspondence, or lack thereof, between these prior works and their current work.

ANSWER. We thank the Reviewer for these two references, which also we missed indeed. They have been included into the manuscript (34 and 35), extensively discussed and referenced.

POINT 3. Figure 1: Legend should be more descriptive. Sequence name starting with the label ‘NRE’ and ending with the label ‘RE’ would be helpful to those not too familiar with GAGs. Also, the orientation should be same (not all atoms are visible in each image which leads to some confusion).

ANSWER. The caption of the figure has been modified. „NRE” and „RE” have been also added to the figure, and the orientation of CS4 from Set 1 has been changed to display all the atoms that have not been visible in the original version of the figure.

POINT 4. Figure A2: Is there a reason why puckering was not restricted using torsional angle restraints? IdoA has been shown to fluctuate between its major conformers on the microsecond timescale (PMID: 20809637). So allowing full ring flexibility on a timescale less than that could be conceived as unrealistic.

ANSWER. We comment on this issue in the revised manuscipt and cite the correspodning work: „The differences in the GlcA and IdoA puckering conformations observed in our study should be, however, interpreted with care. It is known that uronic acids in a monomeric form fluctuate between their major ring pucker conformers on the μs timescale in the simulations with GLYCAM06 force field [53], while each of our simulations had 1 μs length. Since our data present in Figure 9 are obtained without any torsional angle restraint for the rings starting in the same particular conformation, and for each of the residue type within a particular oligosaccharide type there are 12 μs of MD simulation, in total, the observed values could only qualitatively reflect the differences in the puckering conformations that are affected by the limitations of the used force field. Therefore, they cannot be understood strictly in quantitative terms. At the same time, the absence of the torsional restraints allowed to observe that the terminal residues are substantially more flexible in terms of the puckering conformations in comparison to the residues in the middle of the oligosaccharide sequences (Table A1).

POINT 5. For Figure 7, provide chemical structures of the linkages. dp6 has 5 linkages, yet only 'linkage 1' and 'linkage 2' are shown. What is meant by linkage 1 and linkage 2? How do these relate to those of set 1 or set 2?

ANSWER. We have clarified the meaning of the presented data both in the Methods section, in the text and in the captions of Figure 7 and Figure 8.

POINT 6. In Figure 11, intramolecular hydrogen bonding plots are confusing. What do the x and y axes represent? For example, in CS4 dp2, what is the difference between a correlation between 1 and 2, and then between 2 and 2?

ANSWER. We have clarified the meaning of the heatmaps in the captions of Figure 11 and Figure 12.

POINT 7. In general, this reviewer found it difficult to read the x- and y-axes labels and titles in multiple figures (e.g., 2, 3, 4, 5, 6, ….). Similarly, the figures in appendix are also not very readable. These should be greatly improved. A good way to check for format is to print and try to read from a reasonable distance.

ANSWER. The titles and axes labels in figures have been increased in size for convenience of readers.

POINT 8. Line 83: No citations are provided for molecular descriptors. Please add.

ANSWER. We have added the citation to the work [26] in this context: „similarly to other GAG conformational studies [26]”.

POINT 9. Line 96: Better to add PDB IDs rather than just stating PDB.

ANSWER. We have explicitly indicated the PDB IDs of the structures used to build the ligands in the work cited by [33] and replaced the citation for clarity: „The GAG structures used in this study were built from the previously modeled structures of the corresponding GAG molecules [33] originally obtained from the PDB (PDB IDs: 1CS4 and 1HM2).

POINT 10. Lines 86 – 90: Not only is this sentence too long but it is also incomplete. Break it down and clarify.

ANSWER. We have modified and clarified this sentence in the revised manuscript: „We find that the differences between CS4, CS6 and DS can be established by considering a combination of the analyzed dynamic parameters. Our data are in lines with the results obtained for 3-O-sulfated HS variants by Holmes et al. suggesting that the GAG “sulfation code” is a concept which understanding should be established based on the data obtained from the dynamics of these systems.

POINT 11. Lines 159 – 160: Offering a pictorial view of these observations in the form of movie (supplementary?) would be very helpful. How do the authors define conformational flexibility is not clear. It cannot really be based on RMSD alone.

ANSWER. We have added a new supplementary figure (Figure) demonstrating this point.

POINT 12. The authors state (pg 11, ln 206) that a “conformational selection rather than an induced fit mechanism of binding to proteins would be realized”. It is not clear how this conclusion was derived. An explanation should be provided, especially because no analysis has been done in the bound form. In fact, the authors could check multiple computational and non-computational papers published on CS – protein systems and assess whether their hypothesis is correct. Alternatively, the authors could revise it or delete the hypothesis.

ANSWER. We agree with the Reviewer that the original sentence was too strong and not justified enough. We have modified the statement in the revised manuscript: „… suggesting that in terms of glycosidic linkages, there are no significant conformational changes occuring in the process of binding to proteins.

POINT 13. The authors state that convergence was achieved in first few nanoseconds and yet the authors performed simulations for 1 microsecond. Having no restraints on IdoA residues is usually a problem. How were they treated and how was puckering analysed and ascertained to not deviate from what is known?

ANSWER. We have partially clarified this point in our answer to Point 4. In addition, we have corrected our statement on the convergence, respectively: „ … in terms of the RMSD (Figure A1, A2). The complete convergence which would include the ring puckering conformational space of uronic acids requires, however, the MD simulations at the μs timescale [53]”.

POINT 14. Figure A2 legend: Should the ratio be 1C4, 4C1, 2SO and then 1S3, instead of 4C1, 1C4, 2SO, and 1S3?

ANSWER. We have corrected this mistake in Tables A1-A3 legend.

POINT 15. Typographical or grammatical errors should be removed during re-submission. A) Line 1: GAGs is a class (should be ‘are’); B) Line 8: ‘by IdoA acid’ should be ‘in the presence of IdoA instead of ..’; C) Line 16: Same as Line 1; D) Line 57: ‘anvances’ should be ‘advances’; D) Line 63: Cite a reference instead of ‘?’; E) Line 64: ‘predominanly’ should be ‘predominantly’; F) Line 75: fix the charge as superscript; G) Line 190: ‘Oligosacchairide’ should be ‘oligosaccharide’.

ANSWER. All these typos and grammatical errors have been corrected.

Reviewer 2 Report

In this study, a series of MD methods were applied to investigate the differences in structural and dynamic properties of the three oligosaccharide structures CS4, CS6 and DS. In general, each study step was sufficiently logical to draw conclusions that could provide guidance for drug development based on GAG-related pathways. However, some points may need to be complemented.

1. The authors have analysed the MD properties of the three oligosaccharide structures, but more detail on how these differences will affect their binding to other molecules in the pathway is warranted.

2. The authors could provide a more detailed discussion of the conclusions that have been drawn from the dynamic studies of each component.

3.The dynamic simulations are perhaps not fully convincing and the authors could have combined and cited more wet experimental findings in the results and discussion section.

Author Response

POINT 1. The authors have analysed the MD properties of the three oligosaccharide structures, but more detail on how these differences will affect their binding to other molecules in the pathway is warranted.

ANSWER. This has been now additionally highlighted in the revised manuscript in Conclusions: „In particular, the in silico obtained data for these GAGs in terms of their potential protein binding specificity can be interpreted as following. 1. The sulfation in position 6 contributes to the higher flexibility of the GAG in comparison to the sulfation in position 4 and, so could favourize a higher diversity of its possible conformations upon binding, leading to the lower entropic loss upon the formation of a protein-GAG complex and, therefore, higher affinity. At the same time, this higher flexibility of the CS6 could also result in the binding poses more specific than in case of CS4 and DS. 2. There are less available H-bond acceptors in the unbound CS than in the DS meaning that there are more putatively specific patterns of the H-bonding for CS to be expected when bound to the protein. 3. Finally, a more flexible IdoA ring puckering conformational ensemble in DS in comparison to the GlcA one in CS could be the reason for different entropic patterns of these GAGs in the binding. Systematic and rigorous experiments should be performed to find out which thermodynamic trends qualitatively suggested by our data, are dominant in the protein-GAG binding. Isothermal titration calorimetry could be potentially able to answer this question and to verify our findings.

POINT 2. The authors could provide a more detailed discussion of the conclusions that have been drawn from the dynamic studies of each component.

ANSWER. We have elaborated the discussion on the data for each descriptor in the revised manuscript.

POINT 3. The dynamic simulations are perhaps not fully convincing and the authors could have combined and cited more wet experimental findings in the results and discussion section.

ANSWER. Unfortunately, to our knowledge, there are no experimental data for CS and DS that can be directly compared with our results obtained in this work besides the data on the glycosidic linkages from the PDB already discussed in our manuscript. Our previous studies, in which the experimental data available for others GAGs both in bound (PMID: 27496767) and unbound form (PMID: 25490039) were in agreement with the available experimental data. Similarly, other studies (from the research groups of e.g. Desai, Almond and Woods) that were compared with the simulations performed also with GLYCAM06 force field parameters show good agreement between experiments and simulations.

In the manuscript, we additionally comment on the predictive power of the MD-based approaches in terms of their results’ congruence with the experimental data: „It was also shown, with NMR and molecular dynamics (MD)-base techniques} that CS4 and CS6 can bind with significantly different affinities to the same proteins suggesting that it is not exclusively a net charge that drives these interactions .MD simulations have also proved to be successful in reproducing experimental data for unbound CS oligosaccharides. Already in 1999, Kaufmann et al. performed 4 ns MD simulations of the CS4 tetrasaccharide to characterize its glycosidic linkage conformational space, hydration properties and H-bonding.

Round 2

Reviewer 1 Report

The authors have addressed the comments made by this reviewer.